# Arbuscular Mycorrhizal Fungi and Plant Growth-Promoting Rhizobacteria Enhance Soil Key Enzymes, Plant Growth, Seed Yield, and Qualitative Attributes of Guar

Ahmed M. El-Sawah [1], Ali El-Keblawy [2], Dina Fathi Ismail Ali [1], Heba M. Ibrahim [3],
Mohamed A. El-Sheikh [4], Anket Sharma [5], Yousef Alhaj Hamoud [6], Hiba Shaghaleh [7], Marian Brestic [8,9],
Milan Skalicky [8], You-Cai Xiong [10,*] and Mohamed S. Sheteiwy [11,*]

1   Department of Agricultural Microbiology, Faculty of Agriculture, Mansoura University,
    Mansoura 35516, Egypt; ahmedelsawah89@mans.edu.eg (A.M.E.-S.); dfali@mans.edu.eg (D.F.I.A.)
2   Department of Applied Biology, Faculty of Science, University of Sharjah,
    Sharjah P.O. Box 27272, United Arab Emirates; akeblawy@sharjah.ac.ae
3   Department of Botany, Faculty of Agriculture, Mansoura University, Mansoura 35516, Egypt;
    hebaho@mans.edu.eg
4   Botany and Microbiology Department, College of Science, King Saud University, P.O. Box 2455,
    Riyadh 11451, Saudi Arabia; melsheikh@ksu.edu.sa
5   State Key Laboratory of Silviculture, Zhejiang A&F University, Hangzhou 311300, China;
    anketsharma@gmail.com
6   College of Agricultural Science and Engineering, Hohai University, Nanjing 210098, China;
    yousef-hamoud11@hotmail.com
7   Jiangsu Provincial Key Lab for the Chemistry and Utilization of Agro-Forest Biomass, College of Chemical
    Engineering, Nanjing Forestry University, Nanjing 210037, China; hiba-shaghaleh@hotmail.com
8   Department of Botany and Plant Physiology, Faculty of Agrobiology, Food and Natural Resources,
    Czech University of Life Sciences Prague, Kamycka 129, 16500 Prague, Czech Republic;
    marian.brestic@uniag.sk (M.B.); skalicky@af.czu.cz (M.S.)
9   Department of Plant Physiology, Slovak University of Agriculture, 94911 Nitra, Slovakia
10  State Key Laboratory of Grassland Agro-Ecosystems, Institute of Arid Agroecology, School of Life Sciences,
    Lanzhou University, Lanzhou 730000, China
11  Department of Agronomy, Faculty of Agriculture, Mansoura University, Mansoura 35516, Egypt
*   Correspondence: xiongyc@lzu.edu.cn (Y.-C.X.); salahco_2010@mans.edu.eg (M.S.S.)

**Abstract:** Guar is an economically important legume crop that is used for gum production. The clean and sustainable production of guar, especially in newly reclaimed lands, requires biofertilizers that can reduce the use of mineral fertilizers, which have harmful effects on human health and the environment. The present study was conducted to investigate the effects of biofertilizers produced from *Bradyrhizobium* sp., *Bacillus subtilis*, and arbuscular mycorrhizal fungi (AMF), individually or in combinations, on microbial activity, and nutrients of the soils and the guar growth and seed quality and yield. The application of biofertilizers improved shoot length, root length, number of branches, plant dry weight, leaf area index (LAI), chlorophyll content, and nutrient uptake of guar plants compared with the control plants. Moreover, the application with biofertilizers resulted in an obvious increase in seed yield and has improved the total proteins, carbohydrates, fats, starch, and guaran contents in the seeds. Additionally, biofertilizer treatments have improved the soil microbial activity by increasing dehydrogenase, phosphatase, protease, and invertase enzymes. Soil inoculation with the optimized doses of biofertilizers saved about 25% of the chemical fertilizers required for the entire guar growth stages. Our results could serve as a practical strategy for further research into integrated plant-microbe interaction in agriculture.

**Keywords:** *Cyamopsis tetragonoloba*; chemical fertilizers; colonization; seed quality; guaran

## 1. Introduction

Guar (*Cyamopsis tetragonoloba* (L.)) Taub is an important legume crop with high tolerance to drought and salinity stresses [1]. It has the capability to fix atmospheric nitrogen by its effective root nodules, and ultimately, the crop does not require additional nitrogen for the different growth stages [2]. It is cultivated in the world's semiarid areas, including India, Pakistan, and South Africa. In Egypt, guar usually is cultivated for green manuring in newly reclaimed lands, green forage, vegetables, and endospermic gum. A recent study indicated that guar pods' food values are similar to those of the French bean [2]. In recent years, the guar has become an important industrial crop due to its richness of seeds' gum and protein [3]. Moreover, guar gum is a gel-forming galactomannan produced from guar endosperm, a novel agrochemical process. Guaran is a high molecular weight mannose and galactose polymer, which can be used for several industrial purposes [4]. To a great extent, it is used as guar gum powder, an added ingredient in the food, pharmaceuticals, paper, textiles, explosives, oil well drilling, and beauty products industries. This gum powder is also useful in managing various diseases such as diabetes, strong discharges, cardiovascular disease, and carcinoma [5]. Thus, there is a need to study possible ways and technologies to improve the productivity and yield of guar seeds' yield and quality. The use of chemical fertilizers to increase production and yield is not feasible due to their detrimental environmental effects [6–8]. On the other hand, using the biofertilizers for crop production, which depends on the effective free-living soil bacteria and symbiotic nitrogen fixing bacterial in the plant's nodules, is environmentally safe. However, few studies assessed biofertilizers' application in improving guar growth by nitrogen fixation [9,10].

The guar has a low seed yield compared with other legume crops. Hence, improving nitrogen fixation could be a more efficient and appropriate strategy for improving guar seeds' yield. Plant growth-promoting rhizobacteria (PGPR) include a wide range of soil microbes that can provide plants with different benefits such as nitrogen fixation, phosphate solubilization, potassium release, and phytohormones production, in addition to plant growth promotion [11–19]. These microbes can provide essential nutrients to plants, mainly nitrogen, phosphorus, and potassium, which contribute to plant growth promotion and higher yield and minimize the use of chemical fertilizers [20,21]. For example, *Bradyrhizobium* has an important role in improving the plant growth and yield of several species. In a recent study, *B. japonicum* inoculation increased the number of nodules, fresh nodule weight, nitrogen fixation, total nitrogen content, and seed yield in Cluster bean [22]. However, few studies have assessed the impact of *Bradyrhizobium* on the performance of the guar. For example, inoculation of guar plants with various *Bradyrhizobium* spp. strains was an effective approach to increase the effectiveness of guar nodules for fixing atmospheric nitrogen, which increased dry matter, yield, and seed quality of the guar cultivars [11].

Similarly, arbuscular mycorrhizal fungi (AMF) are an integral part of the terrestrial ecosystem and can form a symbiotic relationship with more than 90% of legume plants [23]. AMF symbiosis increases the rate of plant growth by increasing the concentration of P in plant tissues. These fungi regulate the P uptake rate to prolong the availability time of P in soil solution [23,24]. Moreover, AMF improve the physical and chemical properties of the soil, in particular the soil structure. Additionally, AMF symbiosis increased the counts of microorganisms within the rhizosphere and enhanced the activity of soil microbial enzymes [25]. AMF colonization has great potential in the interactions between leguminous plants and soil ecosystems due to its biological and physiological features [26]. These fungi can also improve soil enzymes activities, such as phosphatase and dehydrogenase [7]. Moreover, determination of changes in soil enzymes activities (i.e., acid phosphatase, dehydrogenase, protease, and invertase), mycorrhizal root colonization, and bacterial counts in the rhizosphere of soil could provide a better understanding of the changes in microbial activity in the soil rhizosphere of guar plants treated with biofertilizers.

Hence, the present study aimed at assessing the impacts of inoculation guar plants with *Bradyrhizobium* sp., *Bacillus subtilis*, AMF, and their combinations on soil nutrient availability, the demand for chemical fertilizers, and guar growth, yield, and seed quality.

We hypothesized that inoculation of guar plants with these biofertilizers individually or in combination could increase the soil nutrient availability and reduce the demand for chemical fertilizers. Therefore, these treatments were expected to enhance guar growth, yield, and seed quality and improve the biological properties of the soil. We carried out several physiological and biochemical investigations to evaluate the potential effect of optimized AMF alone or in combinations with various bio inoculants to define the best combination possessing the potential to increase plant growth, nutrient uptake, and seed yield of guar.

## 2. Materials and Methods

### 2.1. Field Experiment and Agricultural Practices

The experimental work was conducted at the Agronomy Farm of Faculty of Agriculture, Mansoura University, Egypt, Mansoura (31.0449° N, 31.3537° E), during the growing seasons in 2018 and 2019 from May to November. The meteorological data for the study area are shown in Figure 1. Clay loam soil was used, soil samples were collected randomly at depths of 0–30 cm from the soil surface before soil preparation, and some physicochemical and biological properties are shown in Table 1. The experiment was performed as a Completely Randomized Block Design (CRBD) with three replicates during both seasons.

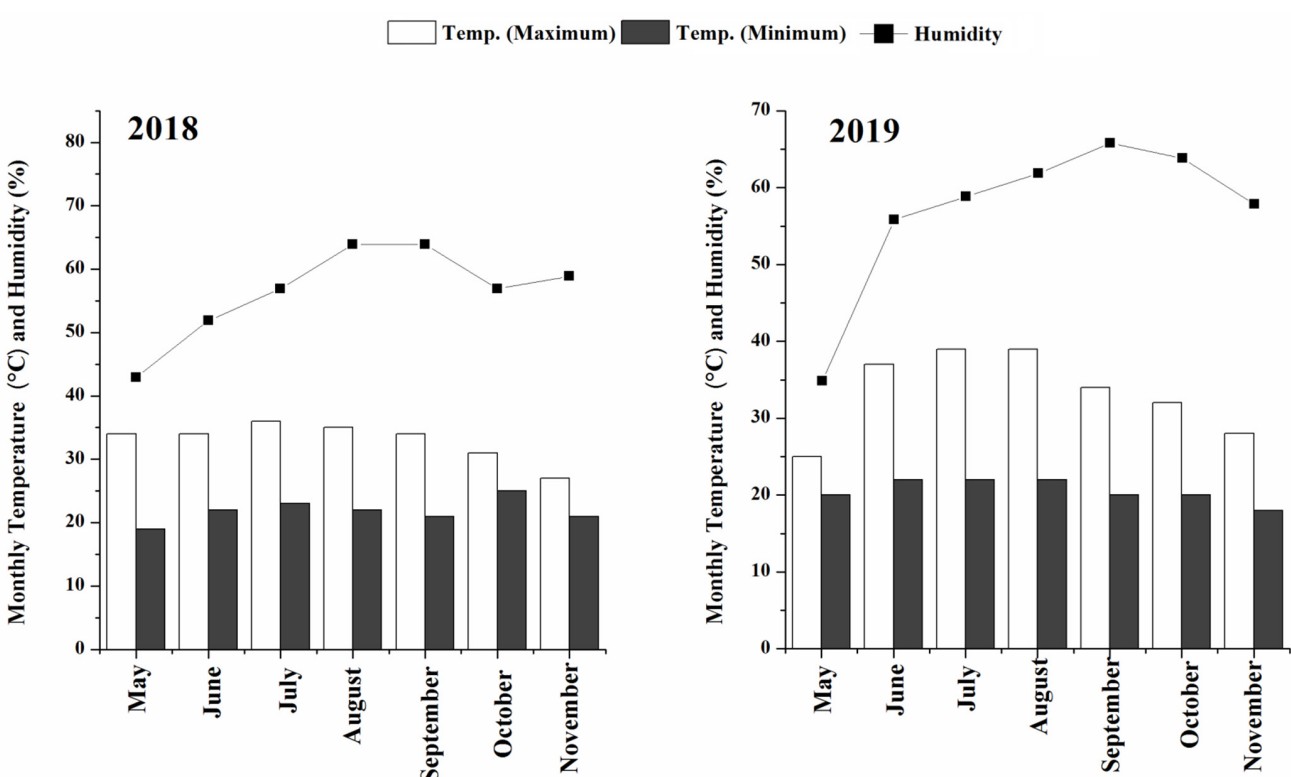

**Figure 1.** Meteorological data of temperature (°C) and relative humidity (%) during 2018 and 2019 growing seasons.

**Table 1.** Physicochemical and biological properties of soil used in the two growing seasons, 2018 and 2019.

| Property | | 2018 | 2019 |
|---|---|---|---|
| Chemical properities | OM% | 2.02 | 1.99 |
| | pH | 7.93 | 8.61 |
| | EC | 2.04 | 2.09 |
| Cations (meq $L^{-1}$) | $Ca^{2+}$ | 11.84 | 9.21 |
| | $Mg^{2+}$ | 6.09 | 4.92 |
| | $Na^+$ | 5.39 | 2.93 |
| | $K^+$ | 0.57 | 0.08 |
| Anions (meq $L^{-1}$) | $CO_3^{2-}$ | 0.00 | 0.00 |
| | $HCO_3^-$ | 3.07 | 1.42 |
| | $Cl^-$ | 10.17 | 10.17 |
| | $SO_4^{2-}$ | 10.66 | 5.56 |
| Available nutrients (mg/kg) | N | 164.00 | 252.0 |
| | P | 9.52 | 9.78 |
| | K | 314.56 | 526.50 |
| Bacterial count | TBC | 6.17 | 6.28 |
| | PSBC | 4.91 | 4.95 |

pH (1:2.5 $H_2O$); O.M. Organic Matter; EC (electrical conductivity $dsm^{-1}$); TBC total bacterial count log (cfu $g^{-1}$ dry soil) and PSBC phosphate solubilizing bacterial count log (cfu $g^{-1}$ dry soil).

Seeds of guar (Baldy), being an important forage crop, were obtained from the Horticulture Department, Faculty of Agriculture, Menia University, Egypt, Menia. The experimental site was occupied with a Gramineae species (*Hordeum vulgare*) prior to the current experiment. The guar seeds were planted on 15 May 2018 and 2019. The guar seeds were prepared for the microbial inoculant according to the study of Gao et al. [7]. Briefly, the seeds were surface sterilized with 1% sodium hypochlorite for 5 min, washed with sterilized water, and then soaked in the bacterial inoculants containing Arabic gum as an adhesive agent for 30 min before planting. However, the mycorrhizal inoculum was inoculated to each hill before planting. Each plot ($3 \times 3.5$ m$^2$) consisted of four 3 m long ridges and row spacing of 50 cm, the hills were 20 cm apart, and 2–3 seeds were planted per hill on both sides of the ridge. Plant thinning was performed three weeks after planting, followed by irrigation, and the subsequent irrigations were applied at 15-day intervals during both growing seasons following the optimized recommendations for guar irrigation under Egyptian conditions, and the weeds were controlled as needed.

The nitrogen fertilizer was added at the rate of 120 kg/ha as urea (46% N) in two equal doses; the first dose was applied after thinning, while the second was applied before the second irrigation. Phosphorus fertilizer was added at a rate of 357 kg/ha as calcium superphosphate (15.5% $P_2O_5$) throughout soil preparation. Potassium fertilizer was added at the rate of 120 kg/ha as potassium sulfate (48% $K_2O$) after thinning together with the first dose of nitrogen fertilizer. The chemical fertilizers were applied according to the recommendations and rules of the Egyptian Ministry of Agriculture, Cairo, Egypt. The biofertilizer treatments received 75% of the recommended doses of NPK.

*2.2. Source and Plant Growth-Promoting Traits of Bacterial Strains*

*Bradyrhizobium* sp. TAL-169 and *Bacillus subtilis* MF497446 were obtained from the Laboratory of Bacteriology, Sakha Agricultural Research Station, Kafr El-Sheikh, Egypt. Phosphate solubilization was measured by determining available soluble phosphate in the Pikovaskya's broth [27] supplemented with tricalcium phosphate as the method of Hemalatha et al. [28]. For Indole acetic acid assay, *Bradyrhizobium* sp. TAL-169 and *Bacillus subtilis* MF497446 were grown in Yeast extract mannitol medium and Nutrient broth (OXOID Ltd., Wade Road, Basingstoke, Hants, RG24 8PW, CM0001, UK), respectively.

L-tryptophan at the rate of (1 g/L) was used as a precursor. IAA was measured according to the method of Ahmad et al. [29].

### 2.3. Bacterial and Mycorrhizal Inocula Preparation

For the preparation of *Bradyrhizobial* inoculum, yeast extract mannitol medium [30] was inoculated with the effective strain (*Bradyrhizobium* sp. TAL-169), then incubated at 30 °C for five days ($1.21 \times 10^9$ cfu/mL). *B. subtilis* MF497446 inoculum was prepared by inoculating nutrient broth medium (OXOID, CM0001, UK), then incubated for three days at 30 °C ($4.03 \times 10^8$ cfu/mL). Guar seeds were inoculated with bacterial cultures according to the study of Gao et al. [7]. The mixture of various AMF spores (*Glomus clarum*, *Glomus mosseae*, and *Gigaspora margarita*) was obtained from the Botany Department, Faculty of Science, Mansoura University, Egypt, Mansoura. The spores of AMF were grown using *Sorghum sudanenses*, L. as a host plant for six months under controlled growing conditions of 20–25 °C, 50% humidity, and day/night period of 16/8 h period. Plants were watered 2–3 times a week and bi-monthly fertilized with non-p fertilizer; phosphorus was added only if signs of deficiency symptoms appeared in host leaves. The trapped soil was used to inoculate mycorrhizal-treated plants according to the study of Gao et al. [7]. Briefly, 5 g of trapped soil and 0.5 g of infected roots of Sudan grass were then inoculated to each hill. The inoculum was placed 5 cm below the surface of the soil before planting.

### 2.4. Staining and Detection of the Levels of Mycorrhizal Colonization

The roots of guar plants were stained with 0.05% trypan blue (SIGMA) according to the method of Hafez et al. [31]. Thirty randomly stained root pieces of each treatment were mounted on slides in lacto-glycerol, squashed, and examined microscopically, and mycorrhizal colonization levels were estimated using Mycocalc software (Wuhan, Hubei, China; https://www2.dijon.inrae.fr/mychintec/Mycocalc-prg/download.html).

### 2.5. Morph-Physiological and Yield of Guar

Four plants were selected randomly from each block on August 15 and used to measure shoot length, root length, number of branches, plant dry weight, LAI, chlorophyll content, and nutrients uptake. At the end of the mature stage, the plants were cut to determine the yield parameters, such as the number of pods per plant, 100-seed weight, and seeds yield per hectare.

### 2.6. Biochemical Analyses

In order to determine the NPK content in the leaves, the oven-dried samples of guar were finely ground, and 0.1 g of samples were moved to the digestion tubes for digestion using $H_2SO_4$ and $H_2O_2$ [32]. Nitrogen was determined using the Kjeldahl method [33]. The phosphorus was determined according to Ashraf et al. [34]. Potassium was determined by a flame photometer [35]. Guaran percentage was determined in the seeds according to the method of Anderson [36]. Crude protein and fat were determined according to the Association of Official Agricultural Chemists (A.O.A.C) [37]. The starch content was determined according to the method of Sene et al. [38]. The phenol-sulfuric method of Dubois et al. [39] was used for total carbohydrate determination, using glucose as a standard.

### 2.7. Enzymatic Assays

The activity of acid phosphatase, dehydrogenase, protease, and invertase were measured in the rhizosphere. Acid phosphatase was measured by the method of disodium phenyl phosphate; triphenyl-tetrazolium chloride (TTC) was used to determine dehydrogenase activity according to the method of Zhang et al. [40]. The activities of protease and invertase were estimated as the method of Xu et al. [25].

### 2.8. Bacterial Count in the Rhizosphere

Rhizosphere soil samples were collected from the rhizosphere soil of four plants. For each treatment, a mixed rhizosphere soil sample was pooled ready for analysis. Total bacteria and phosphate solubilizing bacterial counts were determined in the rhizosphere of guar plants at 45 days after planting. Nutrient agar medium (CM0003, UK) was used for total bacterial counts after three days of incubation at 30 °C, and Pikovskaya medium [27] was used for phosphate-solubilizing bacterial counts; clear zones around the colonies were recorded after five days of incubation at 30 °C.

### 2.9. Statistical Analysis

Each sample was analyzed in triplicate $\pm$ standard deviation (SD). Data were evaluated by one-way analysis of variance (ANOVA) by SPSS v25.0 (SPSS, Inc., Chicago, IL, USA). For separate means, the Duncan multiple range test was used. Significance has been accepted at $p \leq 0.05$.

## 3. Results

### 3.1. Plant Growth Promotion Traits

The results in Figure 2 depicted that *Bradyrhizobium* sp. TAL-169 and *Bacillus subtilis* MF497446 were capable of solubilizing phosphate and effectively producing IAA. The maximum release of soluble phosphorus was 7.60 and 12.40 mg $P_2O_5$/100 mL culture after seven and 21 days of incubation, respectively, by both strains and then decreased with the incubation time ahead Figure 2A. As such, IAA was detected from the first day and reached the maximum level on the fifth day, which was 106.75 and 34.81 µg/mL culture by both strains, respectively, followed by a gradual decrease in IAA production, Figure 2B.

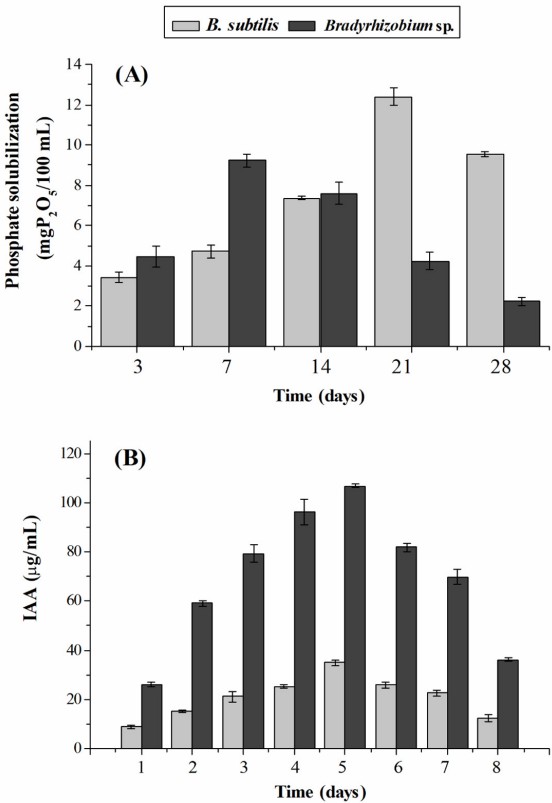

**Figure 2.** Phosphate solubilization. (**A**) Indole acetic acid production (**B**) by *Bradyrhizobium* sp. TAL-169 and *Bacillus subtilis* MF497446.

### 3.2. Morpho-Physiological Traits as Affected by Biofertilizer Treatments

Data on the morpho-physiological traits as affected by biofertilizer treatments are presented in Table 2. It was observed that the biofertilization had a major impact on the morpho-physiological parameters of guar plants during both seasons compared with untreated control plants. Taken together, shoots and root length, number of branches per plant, dry weights, and leaf area, as well as chlorophyll content, were significantly improved by the biofertilization treatments. Nonetheless, the combination treatment significantly resulted in maximum shoot length (223.30 and 214.43 cm) and root length (214.50 and 214.43 cm) followed by Mycorrhiza and *Bradyrhizobium* treatments in both growing seasons, respectively. However, the maximum number of branches per plant (7.33 and 8.66) was recorded with Mycorrhiza treatment, followed by the combination treatment, which was 7.00 and 9.00 in both growing seasons, respectively. Similarly, maximum dry weight (120.96 and 128.63 g) was observed in the combination-treated plants followed by Mycorrhiza-treated plants, which was 103.41 and 106.43 g, and *Bradyrhizobium*-treated plants, which was 96.17 and 100.95 g. Furthermore, the maximum leaf area was observed with the combination treatment (61.84 and 60.15 cm$^2$) followed by Mycorrhiza treatment (53.62 and 55.03 cm$^2$). In addition, the results showed that the combination and *Bradyrhizobium* treatments recorded the highest values of total chlorophyll (49.66 and 50.23 mg L$^{-1}$) and (48.70 and 43.70 mg L$^{-1}$) in both growing seasons, respectively. Such findings indicate that the triple treatment of biofertilizers can improve guar growth by using 75% dose of the required quantity of chemical fertilizers.

**Table 2.** Morpho-physiological traits of guar plants as influenced by bio-fertilizers treatments alone or in combination.

| Year | Treatments | Shoot Length (cm) | Root Length (cm) | Branches/Plant | Dry Weight (g/plant) | LAI (cm$^2$) | Chl. Content (mg L$^{-1}$) |
|---|---|---|---|---|---|---|---|
| 2018 | Control | 183.6 (±1.4) e | 31.4 (±1.8) f | 4.3 (±1.5) c | 74.8 (±2.1) c | 43.1 (±2.0) e f | 31.3 (±1.0) e |
| | *Bradyrhizobium* sp. | 209.6 (±4.2) b–d | 38.4 (±3.5) b–d | 6.3 (±1.1) bc | 96.1 (±4.4) b | 47.5 (±5.5) d–f | 48.7 (±4.1) ab |
| | *B. subtilis* | 201.7 (±4.6) d | 32.1 (±2.2) ef | 5.6 (±0.5) bc | 80.1 (±1.9) c | 41.4 (±1.8) f | 2.7 (±2.6) de |
| | Mycorrhiza | 214.5 (±7.0) a b | 40.2 (±1.8) a–c | 7.3 (±0.5) ab | 103.4 (±0.5) b | 53.6 (±2.8) c d | 41.2 (±3.3) c |
| | Mixture | 223.3 (±6.7) a | 43.6 (±3.3) a | 7.0 (±1.7) ab | 120.9 (±11.5) a | 61.8 (±6.9) a | 49.6 (±2.5) a |
| 2019 | Control | 202.8 (±2.3) c d | 32.5 (±1.1) e f | 4.6 (±1.1) c | 78.9 (±3.5) c | 47.6 (±1.1) d–f | 38.4 (±2.3) cd |
| | *Bradyrhizobium* sp. | 208.0 (±7.2) b–d | 36.9 (±2.9) b–e | 7.0 (±1.0) ab | 100.9 (±11.7) b | 49.0 (±1.0) c–e | 43.7 (±3.1) bc |
| | *B. subtilis* | 202.4 (±2.0) cd | 34.0 (±1.0) d–f | 7.3 (±1.5) ab | 84.5 (±5.5) c | 53.9 (±2.5) b–d | 33.5 (±0.9) de |
| | Mycorrhiza | 211.9 (±8.1) bc | 36.3 (±3.2) c–f | 8.6 (±0.5) a | 106.4 (±4.1) b | 55.0 (±4.1) bc | 42.2 (±6.6) c |
| | Mixture | 214.4 (±1.6) a b | 41.7 (±3.6) ab | 9.0 (±1.3) a | 128.6 (±6.5) a | 60.1 (±2.3) ab | 50.2 (±2.0) a |

Different letters following the data within each column mean significant difference at $p \leq 0.05$. Mixture: (*Bradyrhizobium* sp. + *B. subtilis* + Mycorrhiza); LAI (leaf area index); ChL (Chlorophyll).

### 3.3. Yield and Their Attributes as Affected by Biofertilizer Treatments

Mean data regarding the yield traits as affected by biofertilizer treatments are presented in Table 3. Results showed that the biofertilizer treatments had a considerable effect on the guar yield during both seasons as compared with untreated control plants. The highest number of pods (106.33 and 124.33) and 100 seeds weights (4.28 and 4.58 g) were observed with the combination treatment followed by Mycorrhiza and *Bradyrhizobium* treatments in both growing seasons, respectively (Table 3). A similar trend was also observed by the combined treatment for seed yield weight (2.79 and 2.82 t/ha) with an increase of 30.82% and 17.73% over the control treatment in both growing seasons, respectively (Table 3). It could be stated that the use of triple treatment of biofertilizers combined with 75% NPK increased the yield parameters of guar compared with the control plants.

**Table 3.** Yield properties of guar plants as influenced by bio-fertilizers treatments alone or in combination.

| Year | Treatments | Number of Pods | 100 Seeds Weight (g) | Seeds Yield (ton/ha) |
|---|---|---|---|---|
| 2018 | Control | 92.0 (±7.0) de | 4.02 (±0.07) d | 1.93 (±0.2) c |
| | *Bradyrhizobium* sp. | 95.0 (±2.0) c–e | 4.02 (±0.03) d | 2.29 (±0.2) b |
| | *B. subtilis* | 78.0 (±5.5) f | 4.06 (±0.16) d | 1.91 (±0.2) c |
| | Mycorrhiza | 101.3 (±4.1) bc | 4.11 (±0.08) cd | 2.22 (±0.02) bc |
| | Mixture | 106.3 (±3.0) b | 4.28 (±0.13) bc | 2.79 (±0.01) a |
| 2019 | Control | 90.3 (±4.1) e | 3.92 (±0.02) d | 2.32 (±0.2) b |
| | *Bradyrhizobium* sp. | 100.3 (±5.0) b–d | 4.05 (±0.07) d | 2.48 (±0.20) ab |
| | *B. subtilis* | 89.6 (±3.0) e | 4.03 (±0.13) d | 2.17 (±0.1) bc |
| | Mycorrhiza | 103.6 (±8.6) bc | 4.31 (±0.04) b | 2.47 (±0.1) ab |
| | Mixture | 124.3 (±3.0) a | 4.58 (±0.13) a | 2.82 (±0.02) a |

rDifferent letters following the data within each column mean significant difference at $p < 0.05$. Mixture: (*Bradyrhizobium* sp. + *B. subtilis* + Mycorrhiza).

### 3.4. Effects of Biofertilizers on Nutrient Uptake

To ascertain the state of plant nutrition, the contents of essential plant nutrients, such as nitrogen (N), phosphorus (P), and potassium (K), were evaluated (Figure 3). Mean data with respect to nutrient uptake (N, P, and K) in guar leaves as affected by biofertilizer treatments are presented in Figure 3. Results indicated that the application of biofertilizers significantly improved the content of N, P, and K in guar leaves in both growing seasons as compared with untreated control treatment. The maximum values of N, P, and K contents in leaves of guar plants those inoculated with *Bradyrhizobium* treatment followed by Mycorrhiza treatment. These results suggested that the application of biofertilizers can increase the availability of essential nutrients such as N, P, and K in the soil, which can be translocated to guar through the roots system and thus improve guar growth and yield.

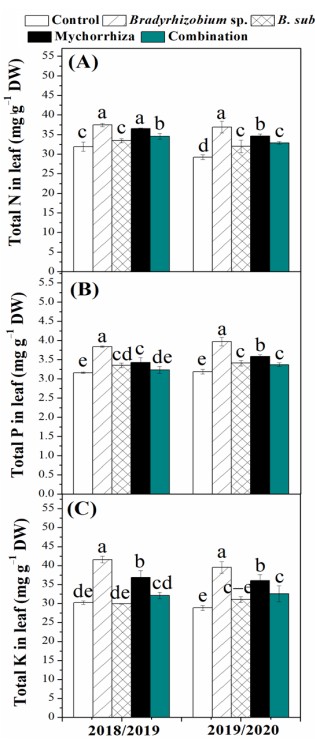

**Figure 3.** Nutrients (N, P, and K) contents in the leaves of guar plants (**A**–**C**) as affected by biofertilizers alone or in combination. Different letters mean significant difference at $p \leq 0.05$.

### 3.5. Effects of Biofertilizers on Seed Quality Properties

The seed quality properties of guar plants as affected by biofertilizer treatments are presented in Table 4. Results showed that total proteins, carbohydrates, fats, starch, and guaran contents were increased due to the combined treatment in which plants were subjected to the joint effect of *Bradyrhizobium*, *B. subtilis*, and mycorrhiza. These seed biochemical constituents were also increased in response to the treatment with either *B. subtilis* or mycorrhiza alone, but with a lower magnitude compared with the combined treatment (Table 4).

**Table 4.** Seed quality properties of guar plants as influenced by bio-fertilizers treatments alone or in combination.

| Year | Treatments | Protein (%) | Carbohydrates (%) | Starch (%) | Fatty Acids (%) | Guaran (%) |
|------|-----------|-------------|-------------------|------------|-----------------|------------|
| 2018 | Control | 26.03 (±0.30) de | 59.13 (±0.23) f | 2.96 (±0.01) e | 1.68 (±0.02) c | 12.03 (±0.03) e |
| | *Bradyrhizobium* sp. | 30.85 (±0.31) a | 60.63 (±0.51) d | 2.88 (±0.02) f | 1.62 (±0.02) d | 12.41 (±0.03) d |
| | *B. subtilis* | 27.63 (±0.53) bc | 61.63 (±0.32) bc | 3.23 (±0.030) c | 1.69 (±0.02) b c | 11.94 (±0.04) ef |
| | Mycorrhiza | 26.86 (±0.24) cd | 59.83 (±0.45) e | 3.13 (±0.06) d | 1.81 (±0.01) a | 11.84 (±0.03) f |
| | Mixture | 30.38 (±0.32) a | 62.73 (±0.37) a | 3.37 (±0.02) a | 1.69 (±0.01) bc | 12.27 (±0.02) d |
| 2019 | Control | 25.83 (±0.25) e | 59.30 (±0.45) e f | 3.00 (±0.01) e | 1.57 (±0.020) e | 13.43 (±0.06) b |
| | *Bradyrhizobium* sp. | 30.48 (±1.15) a | 61.33 (±0.47) c | 2.82 (±0.02) g | 1.60 (±0.03) de | 13.62 (±0.09) a |
| | *B. subtilis* | 28.00 (±0.24) b | 62.03 (±0.25) b | 3.20 (±0.03) c | 1.73 (±0.02) b | 12.57 (±0.15) c |
| | Mycorrhiza | 27.05 (±0.28) c | 60.53 (±0.45) d | 3.14 (±0.03) d | 1.84 (±0.04) a | 12.35 (±0.14) d |
| | Mixture | 30.72 (±0.43) a | 62.86 (±0.25) a | 3.31 (±0.04) b | 1.67 (±0.01) c | 13.52 (±0.07) ab |

Different letters following the data within each column mean significant difference at $p \leq 0.05$. Mixture: (*Bradyrhizobium* sp. + *B. subtilis* + Mycorrhiza).

The enhancing effect of *B. subtilis* treatment was higher than that of mycorrhiza treatment concerning protein, carbohydrates, and starch contents. In contrast, the enhancing effect of mycorrhiza was higher than that of *B. subtilis* concerning fats content. Inoculation with *Bradyrhizobium* increased both protein and carbohydrates contents but decreased starch content (Table 4). On the other hand, fats content was generally unaffected by *Bradyrhizobium* inoculation. When considering the individual effect of the adopted microorganisms, it is evident that *Bradyrhizobium* treatment was more inductive to seed proteins, whereas *B. subtilis* was more inductive to carbohydrates (Table 4). On the other hand, the percentage of fats was the highest in response to mycorrhiza treatment in both seasons. In addition, maximum guaran content (12.41% and 13.62%) was observed with *Bradyrhizobium* treatment followed by the combination treatment (12.27% and 13.52%) in both growing seasons, respectively. Accordingly, these results explored that the biofertilizer treatments could improve the quality of guar seeds and increase the guaran content in the seeds.

### 3.6. Effects of Biofertilizers on Enzyme Activities in the Rhizosphere of the Guar Plant

For further understanding of the beneficial function of biofertilizer treatments to enhance guar growth and yield, the key soil enzymes such as dehydrogenase, phosphatase, protease, and invertase in the rhizosphere of guar plants were studied (Figure 4). The activities of dehydrogenase, phosphatase, protease, and invertase were significantly increased by the application of biofertilizer treatments. Mycorrhizal inoculation actively enhanced dehydrogenase and phosphatase activities in the guar rhizosphere, and the highest dehydrogenase activity was observed with Mycorrhiza treatment followed by the combination treatment in both growing seasons. However, the highest phosphatase activity was observed with Mycorrhiza treatment, followed by *B. subtilis* treatment in both growing seasons. Moreover, *Bradyrhizobium* inoculation significantly enhanced protease and invertase activities. The highest protease and invertase activity were observed with *Bradyrhizobium* treatment followed by Mycorrhiza treatment in both growing seasons. This study suggests that biofertilization could stimulate microbial activity to synthesize

such enzymes to decompose soil organic matter and enrich the availability of nutrients in the rhizosphere of the soil for plant growth promotion.

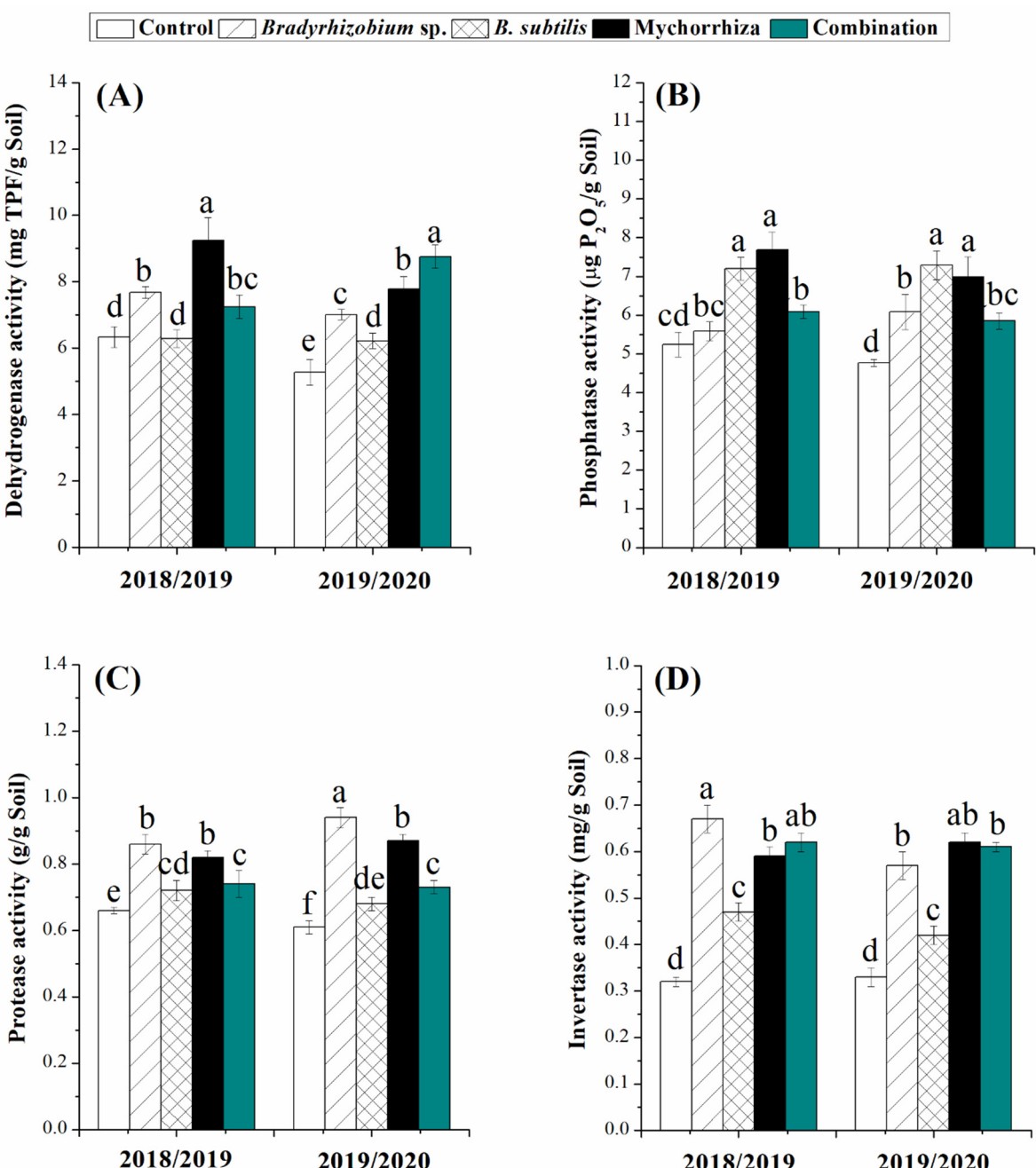

**Figure 4.** Dehydrogenase (**A**), acid phosphatase (**B**), protease (**C**) and invertase (**D**) activities in the rhizosphere of guar plants treated with bio-fertilizers alone or in combination. Different letters mean significant difference at $p \leq 0.05$.

The correlation analysis showed that the number of branches of guar was positively correlated with both dehydrogenase and phosphatase activities (Figure 5A,B) and a similar relationship was also observed with protease and invertase activities (Figure 6A,B), while these enzymes negatively correlated with guaran content (Figure 5C,D and Figure 6C,D). In addition, the correlation analysis showed a significant correlation between the activities of both dehydrogenase and invertase with seed yield (Figures 5E and 6E), and there was a negative correlation in the case of phosphatase and protease activities with seed yield (Figures 5F and 6F).

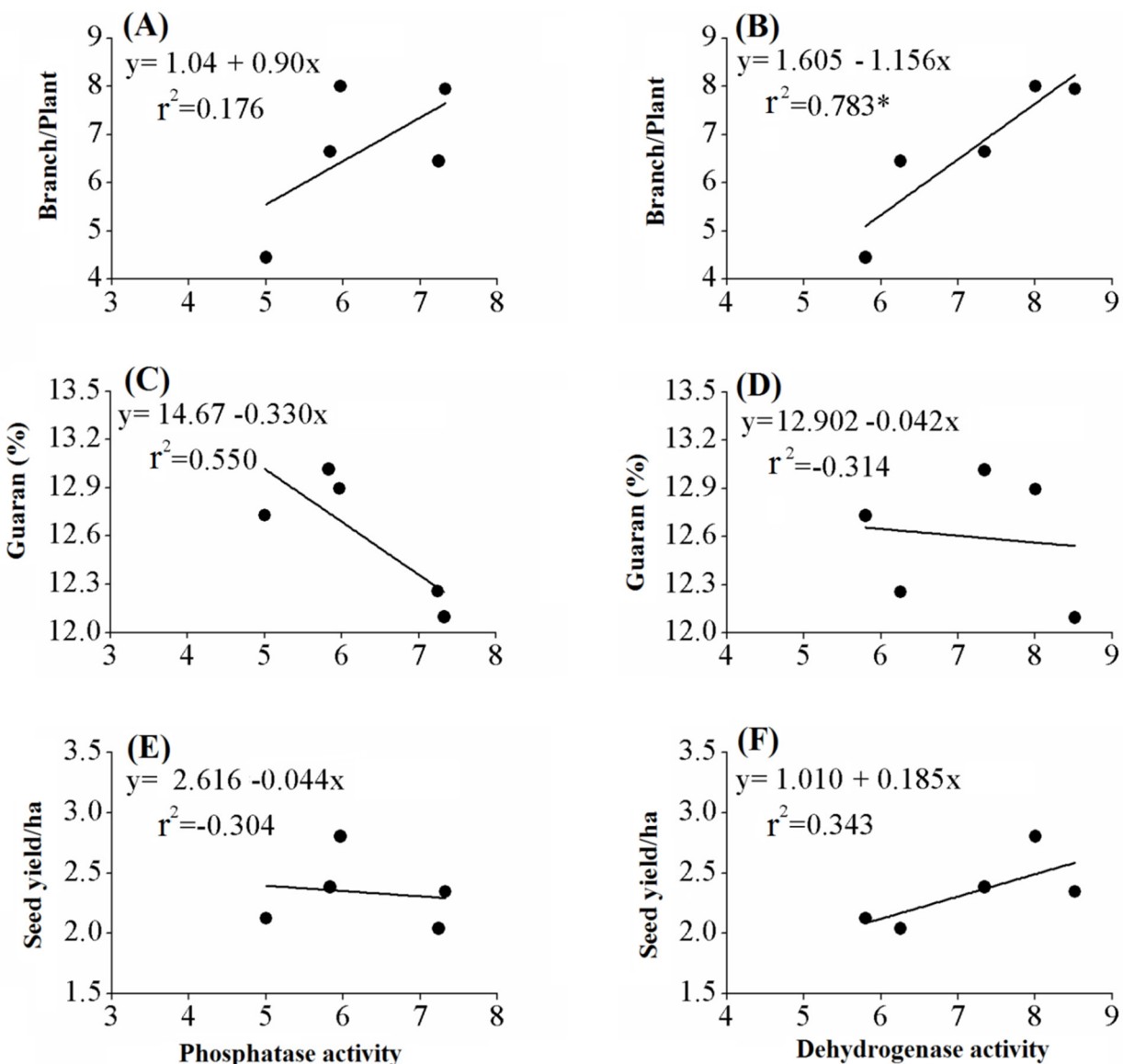

**Figure 5.** Relationship between the mean phosphatase activity and branch/plant (**A**), guaran (**C**), and seed yield (**E**), and the mean dehydrogenase activity and branch/plant (**B**), guaran (**D**), and seed yield (**F**) in guar treated with arbuscular mycorrhizal fungi and plant growth-promoting rhizobacteria. $p \geq 0.01$ (* significant).

### 3.7. Bacterial Counts in the Rhizosphere and Mycorrhizal Colonization in Guar Roots

The effects of biofertilization on bacterial counts in the rhizosphere of guar plants are presented in Table 5. Results revealed that biofertilization enhanced the total bacterial counts and phosphate solubilizing bacteria in the rhizosphere of guar plants treated with biofertilizers as compared with the untreated control treatment. The highest numbers of bacterial counts were recorded in the rhizosphere of the combination-treated plants. This may also be argued that biofertilization has a marked increase in bacterial counts compared with mineral fertilization.

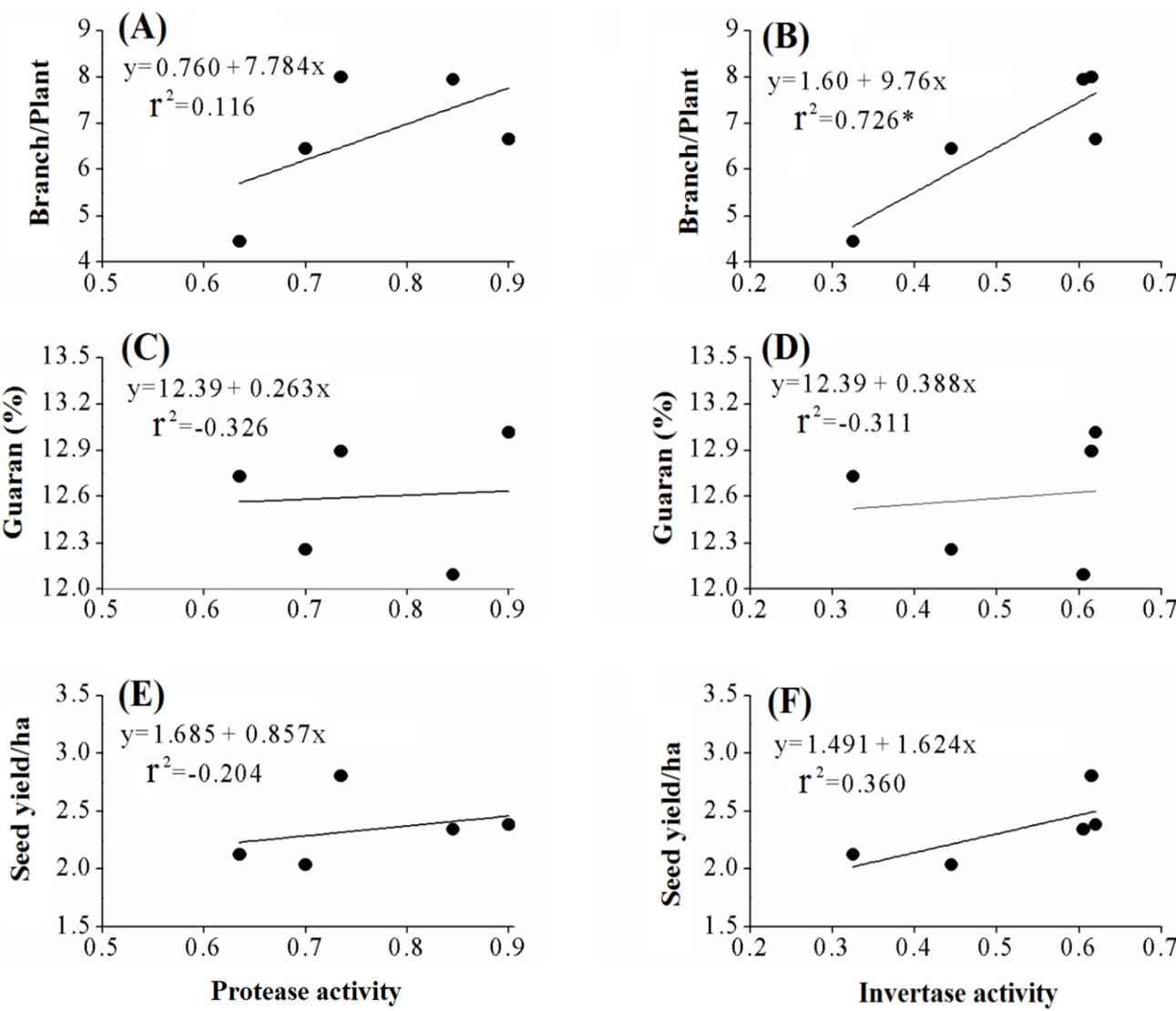

**Figure 6.** Relationship between the mean protease activity and branch/plant (**A**), guaran (**C**), and seed yield (**E**), and the mean invertase activity and branch/plant (**B**), guaran (**D**), and seed yield (**F**) in guar treated with arbuscular mycorrhizal fungi and plant growth-promoting rhizobacteria. $p \geq 0.01$ (* significant).

The levels of mycorrhizal colonization were highly affected by different tested treatments (Table 5). All levels of mycorrhizal colonization were increased by the individual inoculation of mycorrhiza and or with its combination with *Bradyrhizobium* sp. and *B. subtilis.* However, maximum levels of mycorrhizal colonization were observed in the combination treatment in both growing seasons. Structures of arbuscular mycorrhizal fungi (arbuscules, vesicles, and internal hyphae) were observed in trypan blue-stained roots of guar plants (Figure 7). Moreover, no mycorrhizal colonization was observed in the non-mycorrhizal guar plants.

**Table 5.** Bacterial counts in root rhizosphere and mycorrhizal colonization levels in roots of guar treated with biofertilizers alone or in combination.

| Year | Treatments | Bacterial Counts Log (cfu g$^{-1}$ Dry Soil) | | Mycorrhizal Colonization Levels (%) | | | | | |
|---|---|---|---|---|---|---|---|---|---|
| | | 45 DAP | | 45 DAP | | | 90 DAP | | |
| | | Total | P-Solubilizers | F | M | A | F | M | A |
| 2018 | Control | 8.036 f | 5.978 e | – | – | – | – | – | – |
| | *Bradyrhizobium* sp. | 8.204b–d | 6.233 b c | – | – | – | – | – | – |
| | *B. subtilis* | 8.148d e | 6.207 c | – | – | – | – | – | – |
| | Mycorrhiza | 8.243 a–c | 6.276 a–c | 70.00 c | 32.17 c | 15.60 c | 83.33 c | 54.00 c | 37.00 b |
| | Mixture | 8.283 a b | 6.319 a b | 76.67 a | 37.83 a | 17.77 b | 90.00 a | 58.00 a | 39.60 a |
| 2019 | Control | 8.110 e–f | 6.066 d | – | – | – | – | – | – |
| | *Bradyrhizobium* sp. | 8.178 c–e | 6.246 b c | – | – | – | – | – | – |
| | *B. subtilis* | 8.297 a | 6.279 a–c | – | – | – | – | – | – |
| | Mycorrhiza | 8.303 a | 6.287 a–c | 70.00 c | 31.66 d | 16.00 c | 75.00 d | 52.00 d | 35.60 c |
| | Mixture | 8.306 a | 6.349 a | 75.00 b | 34.40 b | 18.66 a | 86.67 b | 56.60 b | 37.40 b |

(–) means no result was detected, different letters following the data within each column means significant difference at $p < 0.05$. F (%): Frequency of root colonization, M (%): Intensity of cortical colonization, and A (%): Frequency of arbuscules. Mixture: (*Bradyrhizobium* sp. + *B. subtilis* + Mycorrhiza). DAP: Days after planting.

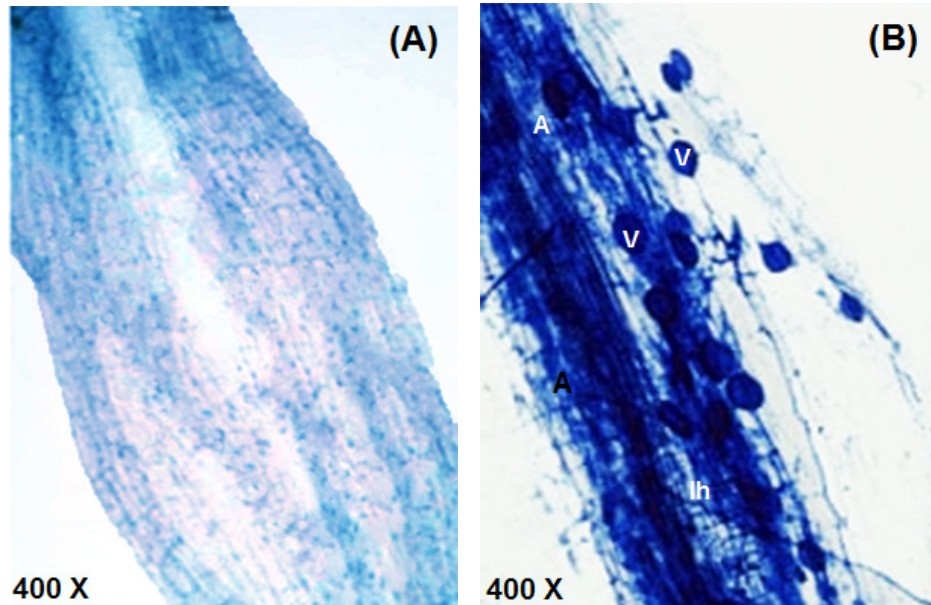

**Figure 7.** Arbuscular mycorrhizal (AM) colonization in the root of guar (*C. tetragonoloba*, L.). Non-mycorrhizal root (**A**), and AMF—colonized root (**B**). Ih—internal hyphae, V—vesicle, and A—arbuscule.

## 4. Discussion

Our results showed that all mycorrhizal colonization levels were increased by the individual inoculation of mycorrhiza or combined with *Bradyrhizobium* sp. and *B. subtilis*. Nevertheless, the mycorrhizal colonization with the combined treatment was slightly higher. These findings are consistent with the results of Abou-aly et al. [41], who found that the combined treatment (AMF + *Azotobacter chroococcum*) resulted in higher mycorrhizal colonization compared with individual mycorrhizal inoculation. Similarly, Juge et al. [42] observed an increase in mycorrhizal colonization in the root system by the combination of AMF and *Bradyrhizobium*, leading to an improvement in water and nutrients uptake by hyphae and an increase of root length density. These findings indicated that the used mycorrhizal species have the ability to colonize guar roots and have the capability to

change specific root system and enhance plant growth by improving phosphorus nutrition and water absorption through their hyphae.

The results also showed that soil inoculation with the biofertilizers activated the microorganisms in the rhizosphere of guar plants. Total bacterial counts and phosphate-solubilizing bacteria were increased in the rhizosphere of plants treated with biofertilizers compared to the untreated control treatment. Increased bacterial counts may be due to nutrient availability in the rhizosphere of biofertilizers-treated plants that provide the needed energy for soil microorganisms to decompose organic matters. These results agree with those obtained by Tewfike [43], who found that inoculation by biofertilizers (*Rhizobium* and *Azospirillum lipoferum*) increased total microbial densities and $CO_2$ elevated in the rhizosphere of guar. A similar study by El-Sawah et al. [14] reported that the biofertilizers increased bacterial counts in the rhizosphere of wheat when applied with both 50% and 75% of the recommended doses of the chemical fertilizers.

Soil key enzymes, mainly dehydrogenase, phosphatase, protease, and invertase, were also activated in the rhizosphere due to increased bacterial counts and the enhancement of microbial activity in guar plants' rhizosphere. In the present study, mycorrhizal inoculation significantly enhanced dehydrogenase activity relative to the other treatments. This could be attributed to the ability of AM fungi to improve the physical and chemical properties of the soil, especially the soil structure. These findings are consistent with Xu et al. [25], who indicated that AM symbiosis increased microorganism counts in the maize rhizosphere and enhanced dehydrogenase activity. Similarly, Gao et al. [7] observed an increase in dehydrogenase activity in the rhizosphere soil of maize plants treated with AM fungi. Our findings revealed that guar rhizosphere inoculated with mycorrhiza and *B. subtilis* had the highest phosphatase activity. The enhanced phosphatase activity may help guar plants to use organic P sources. This finding agrees with that of Ezawa and Yoshida on marigold [44], suggesting that the increase in phosphatase activity could be attributed to mycorrhizal infection. Those authors also stated that AM fungi had a particular phosphatase (MSPase) found only in the mycorrhizal root extract, which is clear evidence of fungal origin. A similar study by Ramesh et al. [45] reported that *Bacillus* spp. can actively produce phosphatase in the rhizosphere of soybean. Xu et al. [25] also reported that maize plants inoculated with biofertilizers, AMF, stimulated the microbial activity in the rhizosphere soil by improving the enzymatic activity. Our results also indicate that biofertilizers significantly enhanced protease and invertase activities, especially in bradyrhizobial and mycorrhizal treatments. Such results are in line with the findings obtained by Xu et al. [36], who observed an increase in protease and invertase activities in the rhizosphere soil of maize treated with AM fungi. Similarly, Zai et al. [46] found that AM fungi increased soil urease and protease activity in the rhizosphere, which increased the nitrogen content of the rhizosphere soil. The increase in soil enzymes can significantly enhance organic matter decomposition and remobilize nutrients in rhizosphere soil.

The effect of *Bradyrhizobial* inoculation on increasing seed protein content is evident from the present study results (Table 4). This effect may be due to *Bradyrhizobial* inoculation-induced compensation of the intrinsically low content and/or low efficiency of native *Bradyrhizobium* in the soil [47,48]. Soil enrichment with *Bradyrhizobium* could boost nodulation and enhance symbiotic nitrogen fixation, culminating in increased plant nitrogen content (Figure 3), thereby enhancing seed protein content [48,49]. Based on this study's findings, total seeds' carbohydrates and starch contents were higher in plants that received the combined treatment, followed by *B. subtilis* treatment during both experimental seasons (Table 4). In this context, previous studies have reported that PGPR treatment, including *B. subtilis*, enhanced photosynthetic pigments [50] and photosynthetic activity [51,52], hence increasing carbohydrates biosynthesis [51]. The enhancement of carbohydrates biosynthesizing machinery in response to PGPR was ascribed to the promotion of these processes by IAA produced by the bacteria [50]. Specifically, *B. subtilis* was reported to enhance the photosynthetic activity in *Arabidopsis* [53] and *Capsicum chinense* [52]. Samaniego-Gámez et al. [52] found that the M9 strain of *Bacillus* increased the

maximum photochemical quantum yield of photosystem II, photochemical quenching, electron transport rate of PSII, PSII operating efficiency, and $CO_2$ assimilation rate. Moreover, *B. subtilis*-induced photosynthesis was attributed to the plant endogenous sugar/abscisic acid signaling [53]. In addition, *B. subtilis*-enhanced photosynthesis could be through the increase of iron uptake and rhizosphere acidification by *B. subtilis*, which facilitates minerals availability [54].

Our results revealed a prominent enhancing effect of AM on seeds fat content (Table 4), which is consistent with the findings of Abd-Allah and Khater on the oily-seeded *Ricinus communis* [55]. This effect may be mediated through the well-established role of AM in facilitating phosphorus availability for the plants [56–59] (Figure 2). As phosphorus is required for lipids biosynthesis, high phosphorus content may enhance lipids biosynthesis, thereby increasing fat content. Previous studies highlighted the functional biology and molecular mechanisms of phosphate uptake by mycorrhizal plants. Hyphal exudates of mycorrhiza were more effective than the plant's root exudates in solubilizing phosphate, contributing to enhanced P availability for mycorrhizal plants [57]. In addition, the extraradical mycelium of AM fungi can secrete phosphatases, which release phosphorus from organic sources [60]. Phosphate taken up by extraradical mycelium is first incorporated into the cytosolic inorganic orthophosphate (pi) pool. The pi surplus is condensed into polyphosphates, which are thought to be the central mediator in supplying pi to the plant [61].

As evident from the data presented in (Table 4), biochemical constituents contributing to seed quality were, mostly, highest in the consortium (combined treatment), in which the plants were benefited from the combined effect of *Bradyrhizobium*, *B. subtilis,* and mycorrhiza. This implies that the microorganisms of the consortium are synergistic in action, i.e., each organism boost the effect of others in the system. One prominent aspect of this synergism is the enhancement of the Rhiz-Legume symbiotic nitrogen fixation through AM-assisted provision of phosphorus. Legumes have an excessive inner phosphorous requirement for their symbiotic nitrogen fixation; so, phosphorous deficiency negatively affects the development of effective nodules and the nodule leghemoglobin content [62]. In this regard, Barea et al. [63] recommend that certain co-operative microbial interactions can be utilized as cost-effective, low-input biotechnology to facilitate sustainable, environmentally-friendly, agro-technological practices.

Our results also showed that inoculation with *Bradyrhizobium* sp. alone or in combination with AMF and *B. subtilis* significantly increased guaran content in guar seeds in both growing seasons. These results are in line with the findings obtained by Abou-Aly [64], who reported that inoculation of guar plants with *Rhizobium* ARC802 combined with *Azospirillium* produced the highest values of seed gum percentage. Similar trends were noticed by Tewfike on guar [43]. Such findings will be of great importance in increasing seed quality properties and the proportion of gum yield.

The current study showed that the application of *Bradyrhizobium* followed by mycorrhizal resulted in an increase of N, P, and K content in guar leaves. The increase in the uptake of these nutrients could be related to the increase in the nitrogen-fixing efficiency by *Bradyrhizobium* [43,63]; N content is usually higher as a direct result of nitrogen fixation. Moreover, the ability of *Bradyrhizobium* to solubilize phosphate (Figure 2) could contribute to the high levels of P and K uptake. Additionally, AMF can absorb phosphorus and other elements through extra radical hyphae and transferring them to the root tissues [23]. Similar observations were reported by Tewfike [43] and Abou-Aly [59], who reported that *Rhizobium* inoculation significantly increased N, P, and K content in guar plants. The increase of P content in guar plants is due to AMF that induces endogenous hormones, contributing to nutrient availability to the plant root. In this regard, Geneca et al. [65] observed that inoculation of pea plants with AMF and *Rhizobium* increased APA and ALA activity. This increase can be attributed to acid and alkaline phosphatase enzymes in the rhizosphere soil produced by AM fungal hyphae. These enzymes help phosphorus

availability to plants, which can be easily absorbed by the colonized root from the soil via mineralization of the bound phosphorus into a soluble form.

In the present study, plant growth and yield of guar were improved by biofertilization. Such improvement could probably be attributed to a significant increase in the availability of soil nitrogen and essential macro and micronutrients, which can enhance guar yield and seeds' quality. Furthermore, the ability of *Bradyrhizobium* and *Bacillus* to produce IAA (Figure 2) could be coupled to improve plant growth and yield. Several studies have reported that biofertilization improves guar growth and yield [11,43,64]. The enhancement in the guar growth might be due to increasing phosphorus uptake by applying AMF, which can increase the population of beneficial microbes in the soil and change the composition of the soil microbial community [3]. Moreover, a recent study reported that the AMF colonization enhanced soil aggregation through extra radical hyphae, which contribute to improving the root system to filter soil water and nutrients. Additionally, soil aggregation could be enhanced through *Trichoderma*, which can secrete several compounds that may improve root colonization, AM spore numbers such as hormones, secondary metabolites, and extracellular enzymes [3,66]. Stancheva et al. [67] reported that AMF fungi and *Rhizobium* increased biomass in the pea plant. Such an increase may be due to no other indigenous microbe competing with inoculated AMF strain, which sprouted rapidly in the soil, resulting in effective inoculation. The extracellular and intracellular hyphae absorb more nutrients for the root surface and ultimately increase plants' growth and biomass.

## 5. Conclusions

From the current study, shoot length, root length, number of branches, plant dry weight, leaf area, chlorophyll content, nutrient uptake, yield, and components of guar plants were significantly affected by the application of biofertilizers and their combination. Activities of soil enzymes such as dehydrogenase, phosphatase, protease, and invertase have also improved in the rhizosphere soil of plants treated with biofertilizers. Increasing soil enzymes in the rhizosphere and the essential nutrients available for the plants increased seed quality by improving the proteins, carbohydrates, starch, fatty acids, and guaran contents. Additionally, the application of biofertilizers reduced chemical fertilizers by 25%. Thus, this research may improve guar production for green forage and seed production for endospermic gum under a lower rate of chemical fertilizers that can be used to achieve greater sustainability of agro-ecosystems. This study suggested that inoculation of guar plants with AMF and *Bradyrhizobium* sp., *Bacillus subtilis* alone, and in different combinations improved plant growth, nutrients' uptake, and significantly improved guar yield. The mixture application of *Bradyrhizobium* sp., *Bacillus subtilis*, and AMF can be recommended to farmers to reduce the use of chemical fertilizers and improving guar growth and yield.

**Author Contributions:** Conceptualization, A.M.E.-S., M.S.S. and D.F.I.A.; methodology, and software; M.S.S., A.M.E.-S., D.F.I.A. and H.M.I.; validation, and data curation, A.M.E.-S. and M.S.S.; formal analysis, A.M.E.-S., M.S.S., Y.A.H. and H.S.; investigation, A.M.E.-S., M.S.S., D.F.I.A. and H.M.I.; resources, A.E.-K., M.A.E.-S., M.B., M.S. and Y.-C.X.; writing—original draft preparation, M.S.S. and A.M.E.-S.; writing—review and editing, A.M.E.-S., M.S.S., H.M.I., D.F.I.A., A.E.-K. and A.S.; visualization, M.A.E.-S., H.M.I., A.E.-K., M.S., Y.A.H., Y.-C.X. and M.B. All authors have read and agreed to the published version of the manuscript.

**Funding:** The authors are grateful for China Post-Doctoral Science Foundation Fund (2019M6617770), and this work was also supported by the project APVV-18-0465.

**Institutional Review Board Statement:** Not applicable.

**Informed Consent Statement:** Not applicable.

**Data Availability Statement:** All the data supporting the findings of this study are included in this article.

**Acknowledgments:** Our deep gratitude is extended to Gamal M. Abdel-Fattah (Botany Department, Faculty of Science, Mansoura University, Egypt) for his sincere help in the section of mycorrhiza. The author M. A. El-Sheikh extends his appreciation to the Researchers Supporting Project Number (RSP-2021/182), King Saud University, Riyadh, Saudi Arabia.

**Conflicts of Interest:** The authors declare that they have no conflict of interest.

## Abbreviations

AMF—Arbuscular Mycorrhizal Fungi; IAA—indole acetic acid; TTC—Triphenyl-tetrazolium chloride; *B. subtilis*, *Bacillus subtilis*; LAI—Leaf Area Index.

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
