# Peer review of "Arbuscular Mycorrhizal Fungi and Plant Growth-Promoting Rhizobacteria Enhance Soil Key Enzymes, Plant Growth, Seed Yield, and Qualitative Attributes of Guar"

_agriculture, doi:10.3390/agriculture11030194_

Round 1

Reviewer 1 Report

The manuscript “Arbuscular mycorrhizal fungi and plant growth-promoting rhizobacteria enhance soil key enzymes, plant growth, seed yield, and qualitative attributes of guar (Cyamopsis tetragonoloba, L.)” provides insight knowledge about practical strategies on the application of biofertilization of C. tetragonoloba L.. However, besides the main objective of this study is clear, the results should be better addressed and its prospects as well.

The introduction is too long, and some information need to be supported by literature.

The methodology section is also confusing regarding any specific analysis. The authors cited all the methods used, but I think a briefly descriptions of the methods are needed. However, there are a lot of English language and sentence errors in overall writeup that cannot be overlooked at this stage. Therefore, editing of English language and style is required by some English mother tongue expert.

Some other major comments are as follows:

The title is too long and, furthermore, I suggest using only “guar” and to put the scientific name in the section keywords.

There are a lot of abbreviations used throughout the manuscript, in this way it is suggested to provide a separate list of abbreviations before the introduction section.

The keywords must be different from title words.

The second major drawback of the manuscript is the results and discussions section which lacks the solid justifications and reasons for the results obtained. Most of the results are too confusing and exaggerated which limits the importance of the manuscript. Moreover, the English language also needs some serious improvements.

Another important thing is the statistics. Why in the figure 5 and 6 the authors correlated the enzyme activities only with some morphological traits and exclude, for example, shoot and root length?

In summary, presented data show an interesting scenario regarding the topics, but discussion of results is very speculative. To sum up, the manuscript can find interest among specialists when these major comments will be taken into account. However, in the present form, I cannot recommend the work for acceptance.

Author Response

­Response to Reviewer 1 Comments

­Dear reviewer:

Thank you for your kind suggestions to improve our manuscript (agriculture-1100220) entitled "Arbuscular mycorrhizal fungi and plant growth-promoting rhizobacteria enhance soil key enzymes, plant growth, seed yield, and qualitative attributes of guar (Cyamopsis tetragonoloba, L.)". The respected reviewer raised some good questions/suggestions for our paper that certainly improved our paper in all aspects. Now, we answer them one by one in the attached file.

Best regards

Reviewer 2 Report

Dear Authors, 

I read your paper and I found that it's written and explained well, also interesting to read. Your approach is sound and I think it may contribute positively to the real world practices by showcasing the impact biofertilizers have on GUAR for fact that it's a field study.

I believe some little changes may improve the presentation of the paper even for the audience. Please see my extended suggestions in PDF file provided. 

Outcome of my review is: Minor changes (with some structural arrangements) and its acceptance once these are made. 

P.S. Please also remember to exclude the alternative text when you save your images inside the word file ( e.g. i see C:\Users\Gana\Desktop\dfdf.tif when i move my cursor on image 1 of pdf file, image 2 asas, image 3 cfcfcf, image 4 wewe and so on) (or you may embed them directly into word file that would eliminate source information). 

Author Response

­Response to Reviewer 2 Comments­

Dear respected editor and reviewers:

Thank you for your kind suggestions to improve our manuscript (agriculture-1100220) entitled "Arbuscular mycorrhizal fungi and plant growth-promoting rhizobacteria enhance soil key enzymes, plant growth, seed yield, and qualitative attributes of guar (Cyamopsis tetragonoloba, L.)". The respected reviewer raised some good questions/suggestions for our paper that certainly improved our paper in all aspects. Now, we answer them one by one as follows.

Best regards

­Response to Reviewer 2 Comments

­Point 2: Line 29: I believe changing the first sentence would be better for readability of the abstract "Guar is an economically important legume crop that is used for gum production”.

Point 4: Lines 56-58: please rephrase and reconsider. Technical details provided in these lines may not be of so much interest for the readers of the agronomical journal.

Point 5: Line 65: cop>crop

Point 6: Line 81: inoculation +of+

Point 7: Line 85: legumes plants > legume plants 

Point 8: line 92: fundi>fungi

Point 9: line 97> inoculation +of+ guar 

Point 10: line 99> treatments “were” _ 

Point 11: I would like to suggest little bit of rearrangement for the readability as the lines 104-107 in my opinion should be put before the hypothesis.

Answer: Dear respected reviewer, many thanks for your concern. We have rearranged this paragraph before the hypothesis.

Point 13: Table 1.  Please add a separate column on the left and move cells (Cations, Anions, Available Nutrients, and Bacterial Counts) in this column with vertical text in order to regroup them in better order. Add thin horizontal lines after EC, K+, SO4, K and PSBC in order to separate them further. 

Point 14: Please follow either one of only 2018 and 2019 seasons throughout the paper, or the format 2018/2019 and 2019/2020.

Point 15: mg/Kg -> mg/kg

Point 16: Please check subscripts of chemical formulas. Some abbreviations used in the tables needs to be explained in table footers e.g. OM%: Organic Matter?  TBC: Total bacterial count? cfu? PSBC:  phosphate solubilizing bacteria count? cfu?  please specify.

Point 17: please unify table 1 and its results in terms decimals used.

Point 18: line 120-> once written in full at the introduction guar now can be referred directly as guar, if you still would like to write the species name C. tetragonoloba should be used.

Point 19: line 129 -> kg not Kg

Point 20: line 134-> biofertilizers treatments

Point 21: lines between 136 and 155-> please add OXOID product numbers, no need to write the first word of the name of the broths or extract in caps (same goes for indole acetic acid). 

Point 22: line 159-> please include details of the software’s producer/developer

Point 23: lines 181-186 -> please detail your rhizosphere sampling methodology. 

Point 24: line 191 -> I assume authors meant statistical significance by P however it should be written in  lowercase italic p. 

Point 25: Line  195-197  ->  misleading  statement.  I  see  highest  levels  in  Fig  2a  for  B  substilis  after  21  days  and  for bradyrhizobium after 7d. Please check your statement. 

Point 26: Fig 2 _ error bars ? the species in the legends are not italic above results

Point 27: Fig 2_ chemical formula P2O5 subscript.

Point 28: line 204 biofertilizers Treatments biofertilizer treatments, authors could also refer these as simply biofertilizers in my opinion, in this way they would avoid also repetitive error of biofertilizers treatments 

Point 29: Tables 2 and 3 -> I think authors did not consider year as separate treatments for the statistical test, and I see that trends of higher results were repetitive for each year, which is completely fine. However, I believe they overlooked the fact that carrying out statistical test by considering both seasons together seems to limit the interpretation of the best performing treatments. I suggest that authors indicate the highest results by making text bold in significantly higher (only the ones with letters a and ab in this case for instance. ( I used a yellow marker here to indicate the higher ones and in this way repeated trend appears clearly in both years)

Point 30: Furthermore, it is not clear to me why significance letters are written in a way that is e-f and d-f. I use another software for the anova and results I have is usually letters written in ef or def (depending on the range). I tried to imagine that SPSS skips the letters in between d and f, therefore d-f is OK but this does not explain e-f. Please clarify.

Point 31: I  would  like  to  suggest  that  authors  consider  updating  tables  by  putting  a  space  between  result,  SD  and significance letter, in this way it’s easier for reader to follow and see them easily. for instance  84.5±5.5c 84.5 (±5.5) c

Point 32: I suggest that for tables 2,3 and 4, 2018 and 2019 are moved vertically into another column to the left of treatments and a thin line between two years would would be added as shown below. 

Point 33: line 278 table 4 same suggestion about formatting the highest ones bold, also goes for this table as evidenced by highlighter the same trend each is easily visible in this way. 

Point 34: Table 2, 3 and 4 -> I believe Ck is the control? I failed to find the full name for this abbreviation within paper. Please reconsider writing simply as Control or include Ck explanation

Point 35: line 300 -> double full stop..

Point 36: line 312 -> please check p value writing and double **

Point 37: line 423-433 -> seems bit out of place and too much detail for the discussion as this paper did not specifically investigated, I would consider the 417-421 as satisfactory for the inorganic orthophosphate (pi) pool c

Point 38: line 459-> AM fungi, AMF.

Point 39: lines 500-503 -> I believe it’s necessary to include some statements about your hypothesis and an answer to your quest to figure out “what was the best combination?” in conclusions as a reference to your lines 102-103 “ to define the best combination possessing the potential to increase plant growth, nutrient uptake, and seed yield of guar”

Best regards,

Reviewer 3 Report

The work presented for review is interesting. The presented problems are current. Agriculture is one of the main perpetrators of unfavorable climate changes, and at the same time it can be one of the tools of climate protection. One of the most important measures is to reduce the use of mineral fertilization in favor of bio-fertilizers. The work is written in understandable language. Contains many interesting research results. However:
- the results should be discussed in the order they are presented,
- the authors should be also limit citing more than 10 years old of literature (there is a lot of older literature).
 Specific comments and questions contained in the text (details in the text).

Author Response

­Response to Reviewer 3 Comments

­Dear respected editor and reviewers:

Thank you for your kind suggestions to improve our manuscript (agriculture-1100220) entitled         "Arbuscular mycorrhizal fungi and plant growth-promoting rhizobacteria enhance soil key enzymes, plant growth, seed yield, and qualitative attributes of guar (Cyamopsis tetragonoloba, L.)". The respected reviewer raised some good questions/suggestions for our paper that certainly improved our paper in all aspects. Now, we answer them one by one as in the attached file.

Best regards

Round 2

Reviewer 1 Report

Dear Authors, many thanks for your kind response. 

Anyway, I'm not convinced from the data. For example, looking to table 5, I sincerely not understand how data varying between 803 and 8.28  CFU/g can results in so significant different.

Please clarify better the statistics. 

Best regards

Author Response

­Dear reviewer:

Thank you very much once again for your kind suggestions to improve our manuscript (agriculture-1100220) entitled "Arbuscular mycorrhizal fungi and plant growth-promoting rhizobacteria enhance soil key enzymes, plant growth, seed yield, and qualitative attributes of guar (Cyamopsis tetragonoloba, L.)". The respected reviewer raised again some good suggestions. Now, we would like to answer them one by one as follows.

Best regards,

Point 1: I'm not convinced from the data. For example, looking to table 5, I sincerely not understand how data varying between 803 and 8.28  CFU/g can results in so significant different. Please clarify better the statistics.

Answer: Dear respected reviewer, many thanks for your concern. We have revised the statistics according to your kind concern and it is ok. The results did not be represented as CFU/g but Log (CFU/g). This is a commonly known method “plate count method” used in recent scientific literature “Hafez, E.; Omara, A.E.D.; Ahmed, A. The Coupling Effects of Plant Growth Promoting Rhizobacteria and Salicylic Acid on Physiological Modifications, Yield Traits, and Productivity of Wheat under Water Deficient Conditions. Agronomy 2019, 9, 524. https://doi.org/10.3390/agronomy9090524”. Also, we are now sincerely explaining to you how results in so significant different with an example as shown in the below Table. You can kindly see from the highlighted columns the wide difference between the two numbers, but we represented it as Log to facilitate the presentation and the statistical analysis. Also, we highlighted Log in Table 5. Please, kindly check Table 5 in the manuscript.

Number of bacterial colonies in the petri dishes from the dilution “106

Mean

Mean after adjustment based on moisture content of soil (= 6.25%) (count/93.75)*100

Log (count)

8.036f

106, 105, 95

102*106= (102000000)

108800000

8.036

8.283ab

170, 190,181

180.333333*106= (180333333)

192355555.2

8.283

Dear respected reviewer, we also have revised the paper regarding English mistakes, and we have tried our best to improve the parts of our manuscript according to your kind suggestions. Please. Thanks a lot once again.

Reviewer 3 Report

Dear authors,

I asked about the type of plant that was grown in this field before the guar.

I still believe that the nitrogen doses used in this crop were too high.

The recommendations of the Ministry are general data.

When establishing the fertilization dose we must know:
-what is the nitrogen demand of the cultivated plant,
-how much mineral nitrogen does the soil in the field contain (how much is left of the pre-crop plant)

-what high of yield we want going to get the yield?.

Sincerely

Author Response

­Dear respected reviewer:

Thank you very much once again for your kind suggestions to improve our manuscript (agriculture-1100220) entitled "Arbuscular mycorrhizal fungi and plant growth-promoting rhizobacteria enhance soil key enzymes, plant growth, seed yield, and qualitative attributes of guar (Cyamopsis tetragonoloba, L.)". The respected reviewer raised again some good suggestions. Now, we would like to answer them one by one as follows.

Best Regards,

­Response to Reviewer 3 Comments:

Point 1: I asked about the type of plant that was grown in this field before the guar.

Answer: Dear respected reviewer, many thanks for your concern. Sorry, we have misunderstood the question. The plant that was grown in this field before the guar is Barley (Hordeum vulgare). Also, we have mentioned this in the methodology section according to your kind suggestion.

Point 2: I still believe that the nitrogen doses used in this crop were too high.

Answer: Dear respected reviewer, many thanks for your kind and valuable observation. The point that you are kindly raised has been previously discussed deeply by our groups, as we have performed several experiments on the same plant species in the same experimental site, which now under processing for the publication. In the previous experiment, we have used different nitrogen doses i.e. 60, 90, 120 and 150 kg/ha, the obtained results showed that 120 kg/ha recorded higher biomass and seed yield as comparing to the lower doses. Based on these results we have selected this dose to be used in the current study. In addition, we tested several plant growth promoting rhizobacteria in particular “Bradyrhizobium spp” to reduce the dose of nitrogen, but there is a problem in nodulation either with the use of high or low nitrogen doses. The used strain “Bradyrhizobium sp. TAL-169” can make nodules on guar roots, but it is still weak in nodulation. However, it has other characteristics “Phosphate solubilization and Indole acetic acid production” (Fig. 2). Moreover, it was shown to record higher biomass in vivo. Also, the strain “Bacillus subtilis MF497446” has the same characteristics as PGPR (Fig. 2) and it can enhance biomass. The mixture treatment containing the two strains in addition to AMF “which can transport elements through their hyphae” leads to a higher biomass and seed yield and can be recommended to farmers to reduce the use of chemical fertilizers by 75% NPK only.

Point 3: The recommendations of the Ministry are general data

Answer: Dear respected reviewer, many thanks for your concern. Although, this dose was recommended by the Ministry of Agriculture, we had carried out pre-experiments to ensure that it was the optimum dose to be used for guar production in this region as mentioned above.

Point 4: When establishing the fertilization dose we must know:
-what is the nitrogen demand of the cultivated plant,
-how much mineral nitrogen does the soil in the field contain (how much is left of the pre-crop plant) -what high of yield we want going to get the yield?.

Answer: Dear respected reviewer, many thanks for your concern. Guar is a newly cultivated crop in Egypt, and as mentioned above, we studied and are still studying the demand of its nutrients. In addition, we analyzed the soil before sowing as shown in Table 1. For the yield, the focus of this study was to increase biomass, seed yield, decrease NPK usage, and to achieve our objectives through the use of biofertilization as shown in Table 2 and 3.